# High Levels of Progesterone Receptor B in MCF-7 Cells Enable Radical Anti-Tumoral and Anti-Estrogenic Effect of Progestin

**DOI:** 10.3390/biomedicines10081860

**Published:** 2022-08-02

**Authors:** Natasa Bajalovic, Yu Zuan Or, Amanda R. E. Woo, Shi Hao Lee, Valerie C. L. Lin

**Affiliations:** School of Biological Sciences, Nanyang Technological University, Singapore 637511, Singapore; natasa.bajalovic.sg@gmail.com (N.B.); yuzuan88@gmail.com (Y.Z.O.); tresors1612@gmail.com (A.R.E.W.); shihao001@e.ntu.edu.sg (S.H.L.)

**Keywords:** MCF-7, progestin, progesterone receptor, antiestrogenic, replicative senescence, breast cancer

## Abstract

The widely reported conflicting effects of progestin on breast cancer suggest that the progesterone receptor (PR) has dual functions depending on the cellular context. Cell models that enable PR to fully express anti-tumoral properties are valuable for the understanding of molecular determinant(s) of the anti-tumoral property. This study evaluated whether the expression of high levels of PR in MCF-7 cells enabled a strong anti-tumoral response to progestin. MCF-7 cells were engineered to overexpress PRB by stable transfection. A single dose of Promegestone (R5020) induced an irreversible cell growth arrest and senescence-associated secretory phenotype in MCF-7 cells with PRB overexpression (MCF-7PRB cells) but had no effect on MCF-7 cells with PRA overexpression. The growth-arresting effect was associated with downregulations of cyclin A2 and B1, CDK2, and CDK4 despite an initial upregulation of cyclin A2 and B1. R5020 also induced an evident activation of Nuclear Factor κB (NF-κB) and upregulation of interleukins IL-1α, IL-1β, and IL-8. Although R5020 caused a significant increase of CD24+CD44+ cell population, R5020-treated MCF-7PRB cells were unable to form tumorspheres and underwent massive apoptosis, which is paradoxically associated with marked downregulations of the pro-apoptotic proteins BID, BAX, PARP, and Caspases 7 and 8, as well as diminution of anti-apoptotic protein BCL-2. Importantly, R5020-activated PRB abolished the effect of estrogen. This intense anti-estrogenic effect was mediated by marked downregulation of ERα and pioneer factor FOXA1, leading to diminished chromatin-associated ERα and FOXA1 and estrogen-induced target gene expression. In conclusion, high levels of agonist-activated PRB in breast cancer cells can be strongly anti-tumoral and anti-estrogenic despite the initial unproductive cell cycle acceleration. Repression of ERα and FOXA1 expression is a major mechanism for the strong anti-estrogenic effect.

## 1. Introduction

Approximately 70% of all breast cancers are hormone-dependent and express estrogen receptor α (ERα) and progesterone receptor (PR). Estrogen-deprivation therapy is the front-line therapy for estrogen-dependent breast cancers, but most patients progressively develop endocrine resistance [1]. PR is an estrogen target gene, commonly used to predict response to anti-estrogen therapy. PR is also known to exert significant influence on the pathology of breast cancer. However, progestin-activated PR in breast cancer can be pro- or anti-tumoral, depending on the context [2,3]. Large clinical trials of hormone replacement therapy (HRT) indicated conclusive association between the use of progestin and breast cancer incidence [4,5,6,7]. Yet, clinical trials using medroxyprogesterone acetate or megesterol acetate have demonstrated improved survival in women with advanced breast cancer [8,9,10,11]. Similarly, laboratory studies also reported both growth-stimulatory and -inhibitory effects in different cell lines, or even in the same cell line. This suggests that progestin-activated PR has dual properties on tumor development, depending on the context. Although progesterone is generally not believed to be carcinogenic in the breast [12,13], it was recently reported to activate GPR126 specifically to promote cell growth through the inhibitory G protein pathway [14]. Knowledge of the molecular determinant(s) for the pro- or anti-tumoral activity of PR will help the identification of biomarkers that predict the anti-tumoral response to progestin.

In laboratory settings, the inherent biology of breast cancer cell lines and levels of PR expression are part of the context. A literature review indicates that progestin can exert a strong growth-inhibitory effect on breast cancer cell lines that express high levels of PR protein. T47D cells were commonly studied because PR is expressed independent of estrogen at relatively high levels. Two independent studies of T47D cells and the derivative overexpressing PRB (T47D-YB) clearly demonstrated that prolonged treatment with R5020 induced growth arrest despite causing initial cell cycle progression [15,16]. In triple-negative breast cancer cells MDA-MB-231 with stable PR overexpression, promegestone (R5020) also strongly and consistently inhibited tumor cell growth both in vitro and in vivo [17,18]. The difference is that R5020 did not elicit the initial cell cycle acceleration in MDA-MD-23 cells as it did in T47D cells. This seems to be correlated to MAPK activation. In T47D cells, R5020 promotes Erk1/2 activation through interaction with ERα [19,20]. In contrast, a single dose of R5020 induced a sustained inhibition of Erk1/2 activation in MDA-MB-231 cells with stable PR expression [21]. MCF-7 cells are the most representative cell model for estrogen-dependent breast cancers, in which PR expression is estrogen-dependent. However, endogenous PR is too low to mediate progestin response in MCF-7 cells in the absence of estrogen. Stable exogenous PR expression with PR levels similar to estrogen-treated MCF-7 cells did not render significant growth-regulatory effects in response to R5020 [22]. It is plausible that higher levels of PR are required to mediate a strong growth-regulatory effect.

Progestin is also known to exert an anti-estrogenic effect in breast cancer cells. The phenomenon was first reported nearly four decades ago in T47D cells and a subline of MCF-7 cells that is tamoxifen-resistant [23]. Several recent studies have described the genomic mechanism of the PR’s anti-estrogenic activity. Progestin-activated PR were found to reprogram ERα cistrome to inhibit estrogen-regulated gene expression and growth [24,25]. On the other hand, there is also evidence of synergy between PR and ERα to promote cell proliferation. It has been reported that PR and ERα synergize to activate c-Src-Erk1/2 cascade [20]. It has also been shown that PRB promotes estradiol- and IGF-1-induced cell proliferation and activation of a subset of estrogen target genes through scafffolding of ERα/PELP1/IGF1R-containing complexes [26]. These studies collectively support the notion that PR also has dual influence on estrogen action.

In the present study, we established MCF-7 sublines with stable expression of high levels of PRB, and the effect of R5020 on the growth characteristics of these cells and its interplay with estrogen/ERα have been elucidated. The study demonstrated that PRB mediated replicative senescence in response to R5020, despite the initial transient cell cycle acceleration. Despite showing an increase of CD24+/CD44+ cell population, R5020-treated MCF-7PRB cells were unable to form tumorspheres and underwent massive apoptosis in Matrigel culture. Furthermore, R5020 completely abolished estrogen’s effect, and this effect was associated with marked downregulation of ERα and FOXA1 expression and their chromatin binding. The study indicates that high levels of PR activity predict a strong anti-tumoral response to progestin through inducing replicative senescence directly and through anti-estrogenic activity.

## 2. Materials and Methods

### 2.1. Cell Culture

The MCF-7 cell line was obtained from ATCC (HB22^TM^). Its identity has been confirmed by short tandem repeat (STR) profiling. MCF-7 cells and its progenies were routinely maintained in Dulbecco’s Modified Eagle’s Medium (DMEM) with phenol red, supplemented with 7.5% fetal calf serum (FCS) (Sigma-Aldrich, St. Louis, MO, USA) and 2 mm L-glutamine. Cells were incubated at 37 °C in a humidified atmosphere of 5% CO_2_ and 95% air.

### 2.2. Establishment of Stable Overexpression of PRB in MCF-7 Cells

MCF-7 cells were transfected with the control vector pcDNA 3.1 (+/hygromycin) (Invitrogen, Carlsbad, CA, USA), or pcDNA 3.1-PRB using Lipofectamine 2000 (Invitrogen, Carlsbad, CA, USA). Stable transfected cells were selected in DMEM containing hygromycin B at 400 µg/mL. Selection of positive clones was based on PRB protein expression by Western Blot analysis. Validation was performed for cells in three consecutive passages before experiments with stable clones were started.

### 2.3. Adenoviral Transduction

Cells were plated in 6-well plates at a density of 2.5 × 10^5^ cells per well for 24 h treatment experiments, and 1 × 10^5^ cells per well for 96 h treatment experiments. At 48 h after seeding, fresh medium was added to the cells. Cells were then loaded with adenovirus AD/ERα at multiplication of infection (MOI) of 5 or AD/PRA at MOI of 1. After 24 h transduction, cells were treated with hormones and experiments were carried out, as described in the relevant sections.

### 2.4. Treatment with Promegestone (R5020) and 17β-Estradiol (E2)

Unless otherwise stated, all experiments involving treatment with the PR agonist R5020 and E2 were carried out in phenol red-free DMEM supplemented with 5% dextran-coated charcoal-treated FCS (DCC-FCS) and 2 mm L-glutamine. The cells were plated and incubated in this medium for 48 h before treatment. R5020 and E2 were reconstituted in 100% ethanol and stored as 10 mM stock solutions. They were used at a concentration of 10 nM with a final ethanol concentration of 0.01%. The quantity of 0.01% ethanol was used as vehicle control.

### 2.5. Cell Growth and Cell Cycle Analysis

To determine the effect of progestin on cell proliferation, 9 × 10^4^ cells were plated in triplicates in the 6-well plates and incubated for 2 days before being treated with R5020 for 4 days. The cells were then collected for counting using a hemocytometer. For cell cycle analysis, a fraction of the cells was stained with propidium iodide (PI) in Vindelov’s cocktail [10 mM Tris-HCl (pH 8), 10 mM NaCl, 50 mg PI/l, 10 mg/l RNaseA, and 0.1% NP40] for 30 min in the dark. The stained cells were analyzed in the BD LSRFortessa™ X-20 flow cytometer (BD Biosciences, Franklin Lakes, NJ, USA) with an excitation wavelength of 488 nm. The resulting histograms were analyzed by the MODFIT LT^™^ program to obtain the distribution of cell populations in the cell cycle phases.

### 2.6. BrdU Incorporation

A total of 9 × 10^4^ cells were plated in 6-well plates for 48 h prior to treatment with 10 nM R5020. After 96 h, BrdU at a final concentration of 20 µM was added to all wells, except for the unlabeled control well. Cells were incubated with BrdU for 30 min before being collected for analysis at 0 h and 16 h. Trypsinized cells were washed with 1× PBS before they were re-suspended in 100 µL of cold PBS. Cells were fixed by being dropped slowly into 5 mL 70% ethanol and incubated overnight at 4 °C. The next day, fixed cells were resuspended in 2 N HCl/Triton X-100 to denature DNA. The samples were neutralized by Na_2_B_4_O_7_, pH 8.5, for 2 min before staining with the anti-BrdU antibody at a concentration of 0.5 µg/mL in PBS containing 1% BSA and 0.5% Tween 20 for 30 min at room temperature. After removing antibodies, cells were resuspended in PBS containing 10 µg/mL of propidium iodide and 10 µg/mL of RNAase A for 30 min in the dark. Samples were analyzed by flow cytometry using the BD LSRFortessa™ X-20 flow cytometer (BD Biosciences, Franklin Lakes, NJ, USA).

### 2.7. Tumorsphere Formation in Matrigel

Cells were first treated with 10 nM R5020 or control vehicle for 72 h. In total, 5000 of the treated cells were seeded on a Lab-Tek’s 8-chamber slide (Thermo Scientific) in 2% Matrigel (Corning Life Sciences) that had been preloaded with 40 μL of 100% Matrigel. The cells were cultured in medium containing 10 nM R5020 or 0.01% ethanol for 14 days and the culture medium was replaced every 3–4 days. The acini structures were observed and imaged on using inverted microscope Olympus IX_71.

### 2.8. Cell Recovery from Matrigel for Quantification

To count live cells in the tumorsphere culture at the end of the 14 days, the plates were incubated with 1 mg/mL Dispase in DMEM supplemented with 2 mM L-glutamine for 1 h at 37 °C. The cells were then collected and pelleted at 250 rcf for 15 min at 4 °C. The pelleted cells were washed once with DPBS, followed by incubation with 0.05% Trypsin at 37 °C for 10 min to obtain single cells. The live cells were quantified based on trypan blue staining.

### 2.9. Effect of R5020 on Cell Growth and DNA Fragmentation in 2% Matrigel

Cells were treated with 10 nM R5020 for 72 h. They were then re-seeded at density of 6250 cells into 24 wells in 2% Matrigel in a culture medium containing 10 nM R5020 or 0.01% ethanol. The cells were cultured in their respective culture treatment for 14 days and the culture medium was replaced every 3–4 days. On day 15, the cells were fixed with 1% paraformaldehyde in PBS for 15 min at room temperature and washed twice with PBS. The cells were then stained with 0.1% crystal violet in 10% ethanol for 15 min at room temperature and washed with deionised water before air-drying. The stained cells were imaged using the HP scanner G4050.

To evaluate the effect of R5020 on apoptosis, the APO-BrdU™ TUNEL Assay was conducted to quantitate DNA fragmentation. Cells treated with 10 nM R5020 for 72 h were re-seeded into 6 wells in 2% Matrigel in a medium containing R5020 or 0.01% ethanol. After 48 h, cells were collected for the analysis of DNA fragmentation using the APO-BrdU™ TUNEL Assay Kit based on the manufacturer’s protocol (Thermo Fisher Scientific, Waltham, MA, USA).

### 2.10. Protein Lysate Collection and Western Blotting Analysis

Cells were lysed with a cold lysis buffer, as reported [27]. The supernatant was collected after centrifugation and proteins in total cell lysates were resolved by SDS-PAGE electrophoresis before being transferred onto a nitrocellulose membrane or polyvinylidene difluoride (PVDF) membrane. Following blocking with 5% skimmed milk or 2.5% BSA, the membranes were probed with primary antibodies overnight at 4 °C. The primary antibodies used in the experiments are: H190 Total PR (Santa Cruz Biotechnology Inc., Dallas, TX, USA, LOT number sc-7208), ERα (Santa Cruz Biotechnology Inc., Dallas, TX, USA, LOT number sc-8002), p21 (Cell Signaling Technology, Danvers, MA, USA, LOT number #2946), p53 (Cell Signaling Technology, Danvers, MA, USA, LOT number #9282), β-actin (Santa Cruz Biotechnology Inc., Dallas, TX, USA, LOT number sc-47778), GAPDH (Ambion, USA, LOT number AM4300), CDK2 (Santa Cruz Biotechnology Inc., Dallas, TX, USA, LOT number sc-6248), CDK4 (Santa Cruz Biotechnology Inc., Dallas, TX, USA, LOT number sc-23896), Cyclin A2 (Cell Signaling Technology, Danvers, MA, USA, LOT number #4656), Cyclin B1 (Cell Signaling Technology, Danvers, MA, USA, LOT number #12231), Cyclin E1 (Cell Signaling Technology, Danvers, MA, USA, LOT number #4129), Cyclin D1 (Cell Signaling Technology, Danvers, MA, USA, LOT number #2978), pan IkBa (Cell Signaling Technology, Danvers, MA, USA, LOT number #4814), IkBa-pS32/36 (Cell Signaling Technology, Danvers, MA, USA, LOT number #9256), p65 (Cell Signaling Technology, Danvers, MA, USA, LOT number #8242), p65-pS536 (Cell Signaling Technology, Danvers, MA, USA, LOT number #3033), RB (Cell Signaling Technology, Danvers, MA, USA, LOT number #9309), RB-pS780 (Cell Signaling Technology, Danvers, MA, USA, LOT number #9307), Caspase 7 (Cell Signaling Technology, Danvers, MA, USA, LOT number #9492), Caspase 8 (Cell Signaling Technology, Danvers, MA, USA, LOT number #4790), PARP (Cell Signaling Technology, Danvers, MA, USA, LOT number #9542), BID (R&D Systems, MN, USA, LOT number AF860), BAX (Cell Signaling Technology, Danvers, MA, USA, LOT number #2772), BCL-2 (Santa Cruz Biotechnology Inc., Dallas, TX, USA, LOT number sc-7382), LC3B (Cell Signaling Technology, Danvers, MA, USA, LOT number #2775, FOXA1 (Cell Signaling Technology, Danvers, MA, USA, LOT number #58613), EZH2 (Cell Signaling Technology, Danvers, MA, USA, LOT number #5246), and H3 (Cell Signaling Technology, Danvers, MA, USA, LOT number #14269). The primary antibodies were used in dilution 1:1000, with exception of GAPDH, which was diluted at 1:20000. The secondary antibodies, anti-mouse (1:1000) and anti-rabbit (1:2000) (GE Healthcare, Chicago, IL, USA), conjugated with HRP, were used according to the class of rimary antibodies. Membranes were washed after incubation with primary and secondary antibodies and the proteins of interest were detected by Millipore’s chemiluminescence reagent and exposure onto X-ray film.

### 2.11. RNA Extraction and Gene Expression Analysis

The total RNA was extracted with TRIzol reagent (Life Technologies), based on the manufacturer’s instructions. RNA was reverse-transcribed to cDNA using the qScript cDNA Synthesis kit (Quanta BioSciences), according to the manufacturer’s instructions. qRT-PCR was performed by using the SYBR green master mix (KAPA Biosystems) and an ABI QuantStudio 6 Detection System (Applied Biosystems). The qRT-PCR for each gene was performed in triplicates. The fold changes were calculated by normalizing CT values of the gene with CT values of 36B4 gene. Primers used for qRT-PCR are listed in Table 1.

### 2.12. Flow Cytometry Analysis of CD44+ and CD24+ Population

Cells were plated at a density 9 × 10^4^ in 6-well plates for 48 h prior to treatment with 10 nM R5020 or 0.01% ethanol. Cells were collected at 96 h after treatment with 0.05% trypsin/0.025% EDTA and resuspended in PBS containing 2% FBS and then stained with anti-CD44-PE and anti-CD24-FITC, according to the manufacturer’s instructions. The samples were then washed by PBS and finally re-suspended in PBS containing 2% FBS. Flow cytometry analysis was performed on a BD LSRFortessa™ X-20 flow cytometer (BD Biosciences, Franklin Lakes, NJ, USA).

### 2.13. Detection of Chromatin-Bound ERα and FOXA1 by Cell Fractionation

Cells were treated with or without R5020 for 24 or 96 h before they were collected and re-suspended in buffer C1 (10 mM Hepes pH 7.9, 1.5 mM MgCl_2_, 10 mM KCl, 0.1% NP40, 10 mM DTT, 5 µg/mL pepstatin A, 5 µg/mL leupeptin, and 2 µg/mL aprotinin). After incubation on ice for 10 min, cells were passed through a 21G syringe 10 times. The supernatant after centrifugation at 1000× *g* was designated as the cytoplasmic fraction, and the pellet contained the nuclei. The pellet was washed once in buffer C1, re-suspended in buffer N1 (10 mM Hepes pH 7.9, 1.5 mM MgCl_2_, 10 mM KCl, 0.5% NP40, 10 mM DTT, 5 µg/mL pepstatin A, 5 µg/mL leupeptin, and 2 µg/mL aprotinin), passed through a 29G syringe, and centrifuged at 12,000× *g* for 10 min to obtain the nuclear soluble fraction in the supernatant. The pellet was re-suspended in buffer N2 (10 mM Hepes pH 7.9, 1.5 mM MgCl_2_, 0.4 M KCl, 0.3 M sucrose, 0.5% NP40, 10 mM DTT, 5 µg/mL pepstatin A, 5 µg/mL leupeptin, and 2 µg/mL aprotinin) and passed through a 29G syringe 10 times and spun down at 20,000× *g* for 15 min to obtain the chromatin-bound fraction.

### 2.14. Statistical Analysis

Data are expressed as mean ± SEM (standard error of the mean). The statistical differences between the treated group and the controls were determined by unpaired Student’s *t*-test using the program on a GraphPad Prism 6. The degree of statistical significance is indicated with asterisks (* *p*< 0.05, ** *p* < 0.01, *** *p* < 0.001, **** *p* < 0.001).

## 3. Results

### 3.1. R5020 Induces Replicative Senescence in MCF-7 Cells Expressing High Levels of PRB

Stable PR overexpression in MCF-7 cells was generated with the transfection of PRB cDNA in pcDNA3.1 plasmid. Three clones of control vector-transfected cells and PRB-transfected cells were characterized. Endogenous PR was not detectable under the experimental conditions in the estrogen-deprived medium for vector-transfected cells (MCF-7C) (Figure 1A). In cells transfected with PRB cDNA (MCF-7PRB), PRB is expressed as the main isoform. PRA is expressed from the second ATG of PRB cDNA at a much lower level compared to PRB. Although the protein levels of ERα were similar in untreated MCF-7C and MCF-7PRB clones, R5020 treatment for 24 h markedly reduced ERα protein in MCF-7PRB cells. This is consistent with previous reports that progestin down-regulates the expression of ERα [28,29], but the magnitude of the downregulation is far greater with higher levels of PRB protein. Remarkably, R5020 treatment for 96 h reduced the cell number of MCF-7PRB by ~50% in all three clones but had no detectable effect in the three clones of vector-transfected control MCF-7C cells (Figure 1B). Since all three clones of MCF-7C and MCF-7PRB cells behave similarly in response to R5020 in terms of growth, subsequent studies focused on one clone of MCF-7C, and one or two clones (PRB2 and PRB3) of MCF-7PRB cells, depending on the experiments.

Cellular senescence is defined by a loss of replicative capacity in previously proliferating cells [30]. Senescent cells are characterized by features such as morphological changes, irreversible cell cycle arrest, and an increase of senescence-associated β-galactosidase (Saβ-gal) activity. It was found that R5020-treated MCF-7PRB cells were flattened with a large surface area and enlarged nuclei, characteristic of senescent cells (Figure 1C). To confirm that R5020 induced a loss of replicative capacity in MCF-7PRB cells, growth was evaluated after the R5020 washout. In this experiment, MCF-7PRB cells were given one dose of R5020 or a control vehicle (0.01% Ethanol). After 96 h, the medium was removed and cells were washed with PBS before fresh medium without R5020 was added. Consistently, R5020 treatment for 96 h reduced cell numbers by approximately 40-50% (Figure 1D, left panel). Cells cultured for additional 96 h after R5020 washout remain growth-arrested with the cell numbers being approximately 25% of the vehicle-treated controls in both clones of MCF-7PRB (Figure 1D, right panel). In contrast, the number of vehicle-treated cells tripled over the second 96 h in culture.

To further determine that R5020 prevents DNA replication in MCF-7PRB cells, a 5-Bromo-2′-deoxyuridine (BrdU) incorporation assay was carried out. Cells treated with R5020 for 96 h were pulse-labeled with 20 μm BrdU for 30 min and collected immediately (0 h) or 16 h later for FACS analysis (Figure 1E). At 0 h, ~19% of MCF-7C cells were labelled with BrdU in the absence or presence of R5020. In contrast, only 3% of R5020-treated MCF-7PRB cells were labelled with BrdU as compared to 23% in vehicle-treated cells (Figure 1F). At 16 h later, the number of BrdU-labeled cells in vehicle-treated MCF-7PRB cells was increased from 22% to 30%, an indication of cell replication. This is also evident from the progression of BrdU-labeled cells through the cell cycle from the 0 h to 16-h time points (Figure 1E). In contrast, BrdU-labeling in R5020-treated cells at 16 h is similar to that at the 0 h time point, indicative of a lack of cell division (Figure 1F). Collectively, the study provided strong evidence of progestin-induced replicative senescence in MCF-7 cells with high levels of PRB.

### 3.2. PRA Exerts No Effect on Cell Growth in Response to R5020

PRA and PRB are known to exert differential gene regulation and modulate different reproductive functions [31]. PRA also trans-represses the activity of PRB in transcription assays [32]. To test the effect of PRA, PRA was overexpressed in MCF-7C and MCF-7PRB cells through adenoviral vector-mediated gene delivery [22]. PRA overexpression did not evidently reduce ERα and did not affect PRB-mediated ERα downregulation in response to R5020 (Figure 1G). R5020 also did not affect cell proliferation in MCF-7C cells with transient PRA overexpression (Figure 1H). This is consistent with the observation in T47D cells [33]. Although PRA levels were 3-4 times higher than PRB in MCF-7PRB cells, there was no demonstrated effect on PRB-mediated growth inhibition in MCF-7PRB cells in response to R5020.

### 3.3. Progestin Exerts Biphasic Effect on Cell Cycle Regulators

In T47D cells overexpressing PRB, progestin is known to exert a biphasic effect on cell cycle regulators that elicited cell cycle progression in the first mitotic cell cycle, and subsequently, cell cycle arrest [15]. Similarly, R5020 increased the levels of cyclin A2 and cyclin B1 at 24 h after treatment but reduced them by 96 h (Figure 2A,B). These changes were associated with the increase and decrease in S-phase cells at 24 h and 96 h, respectively (Figure 2C; representative histograms of different phases of the cell cycle are shown in Appendix A). There is also a significant increase of G2/M phase cells after 96h R5020 treatment. Thus, progestin also exerts biphasic effects on cell cycle progression in MCF-7 cells with high levels of PRB. At the 96 h time point, R5020 caused marked reductions in cyclin-dependent kinases 2 and 4 (CDK2 and CDK4) (Figure 2A,B), which are important for S-phase entry. However, the levels of the p21 (CIP1/WAF1) protein, an important cyclin-dependent kinase inhibitor, were not affected by R5020 treatment.

Intriguingly, R5020 treatment also caused a notable increase of cyclin E1 protein at both the 24 h and 96 h time points (Figure 2A,B). This occurs despite a reduction at the mRNA level (data not shown). Cyclin E-dependent CDK2 activity activates E2F for S-phase entry via phosphorylation and degradation of retinoblastoma protein (RB) [34]. It was found that R5020 induced an increase of cyclin E1 at 96 h and is associated with a decrease of total RB protein (Figure 2A,B) suggestive of RB degradation. However, the R5020-induced increase of cyclin E is not associated with an increase of c-Myc expression (data not shown), a well-known E2F target gene. This suggests that the R5020-induced increase of cyclin E is not associated with functional activation of E2F.

### 3.4. R5020-Induced Replicative Senescence in MCF-7PRB Cells Is Associated with Senescence-Associated Secretory Phenotype (SASP)

Senescent cells are known to secrete a myriad of proteins that influence the behaviors of neighboring cells, and the phenomenon is known as senescence-associated secretory phenotype (SASP) [35]. The secreted factors in SASP are mostly soluble factors. The pro-inflammatory cytokines interleukins (IL), IL-1α, IL-1β, IL-6, and IL-8 are the most commonly reported cytokines of SASP [36]. Particularly, recombinant IL-1α has been found to inhibit cell growth and induces cell cycle arrest in MCF-7 cells [37,38]. We found that there was a marked increase in the expression of *IL-1α* and *IL-1β* in response to R5020 after both 24 h and 96 h treatment (Figure 3A,B). Interestingly, *IL-8* was not up-regulated by R5020 in MCF-7PRB cells at 24 h (Figure 3A) but markedly up-regulated after 96 h treatment (Figure 3B). It has been reported in fibroblasts that membrane-bound IL-1α is critical for maintaining senescence-associated secretion of IL-6/IL-8 in a positive feedback loop [39]. Hence, R5020-induced *IL-1α* may be responsible for the increased *IL-8* expression at later time point, reinforcing the SASP phenotype. The *IL-6* expression level was too low to be reliably evaluated.

NF-κB is regarded as a master regulator of SASP and p65 is a major transcription factor that accumulates on chromatin of senescent cells [40]. p65 activity is controlled by the NF-κB inhibitor α (IκBα) and IκB kinase. Association of IκBα with NF-κB inactivates NF-kB by masking their nuclear localization signals so as to keep them inactive in the cytoplasm [41]. Phosphorylation of IκBα by IκB kinase results in the dissociation of IκBα, allowing the nuclear translocation and activation of NF-κB. p65 phosphorylation at S536 renders NF-κB activation independent of IκBα [42]. Surprisingly, both IκBα (*NFKB1A*) mRNA and protein levels were upregulated by R5020 at 24 h and 96 h (Figure 3A,B). However, there were also marked increases of IκBα phosphorylation at all the time points (24, 30, 46, and 96 h) evaluated (Figure 3C,D), which signals IκBα dissociation from NF-κB and degradation. Note that phospho-IκBα in MCF-7C cells and vehicle-treated MCF-7PRB cells were not detectable, suggesting that the basal phosphorylation of IκBα in MCF-7 cells under the experimental condition is very low. Consistent with increased IκBα phosphorylation, phospho-p65 (S536) levels relative to total p65 were increased by R5020 in MCF-7PRB cells and statistically significant at the 24 and 30 h time points (Figure 3C,D). Taken together, R5020-induced phosphorylation of both IκBα and p65 results in the activation of the NF-κB pathway, leading to SASP with increased expression of cytokines, such as *IL-1α*, *IL-1β,* and *IL-8,* in MCF-7PRB cells.

### 3.5. R5020 Enhanced CD24+CD44+ Population in MCF-7PRB Cells

CD44 is widely used as a cancer stem cell marker. Progestin has been reported to increase the population of CD44+ cells in T47D and MCF-7 cell models [43]. It is therefore important to evaluate whether R5020 influences cancer stem cell properties whilst it induces replicative senescence. Consistent with a previous report [44], 99% of untreated MCF-7 cells in steroid hormone-deprived medium are CD24+CD44- (Figure 4A). R5020 has no effect on the distribution of cells expressing CD24 or CD44 in MCF-7C cells. However, R5020 markedly increased CD44+ populations. consequently, there were 35% and 41% of CD24+CD44+ cells, in R5020-treated PRB2 and PRB3 cells, respectively, as compared with ~1% in untreated cells (Figure 4A). Thus, R5020 could enhance cell population expressing cancer stem cell markers while inhibiting cell growth.

### 3.6. R5020 Inhibits Tumorsphere Formation and Induces Massive Apoptosis of MCF-7PRB Cells in Matrigel Culture

Mammosphere formation assay is commonly used to measure mammary stem cell activity. The same assay has also been used to evaluate the stemness and aggressiveness of cancer cells and this is commonly known as a tumorsphere assay. This three-dimensional (3D) cell culture in Matrigel imitates the in vivo cell organization because 3D interaction between cells and extracellular matrix is crucial for cell activity. The assay is thus useful for evaluating the effect of R5020 on cancer cell stemness. Cells were treated with R5020 for 72 h before being re-suspended in 2% Matrigel and laid onto 100% Matrigel. It was found that MCF-7C cells formed typical tumorspheres with or without R5020 treatment (Figure 4B). Whilst vehicle-treated MCF-7PRB cells also formed tumorspheres similarly, R5020-treated MCF-7PRB cells formed no tumorspheres (Figure 4B). To ascertain that the small clusters of cells in R5020-treated cells are indeed non-viable cells, all the cells in 100% Matrigel were collected and counted. It is evident that there were very few viable cells in R5020-treated MCF-7PRB cells (Figure 4C).

To ensure that equal number of cells were plated in the tumorsphere assay, cells from the same preparation were also plated onto a 24-well plate in 2% Matrigel for observation. Notably, very few cells remained in R5020-treated wells at the end of the tumorsphere experiment as the dead cells were removed during the routine medium change every 3 days (Figure 4D). To verify that these R5020-treated MCF-7PRB cells died by apoptosis, DNA fragmentation was evaluated by a TUNEL assay, which showed that ~50% and 35% of R5020-treated PRB2 and PRB3 cells, respectively, underwent DNA fragmentation at 48 hours’ post-plating in 2% Matrigel (Figure 4E,F). We can conclude from these data that R5020 does not enhance cancer cell stemness, and R5020-treated MCF-7PRB cells are unlikely to form tumors or establish metastasis in vivo.

We subsequently explored molecular events that are responsible for progestin-induced cell death in 2% Matrigel. Surprisingly, R5020-induced massive apoptosis was associated with marked decreases of the pro-apoptotic proteins p53, p21, BID, BAX, and poly ADP ribose polymerase (PARP) (Figure 4G). Since MCF-7 lacks Caspase 3, we probed for protein levels of Caspase 7 and 8. Both non-cleaved and cleaved forms of Caspase 7 were also reduced markedly in R5020-treated MCF-7PRB cells. Cleaved Caspase 8 was not detectable but there is a decrease of non-cleaved Caspase 8 in R5020-treated PRB cells. On the other hand, BCL-2, a very important anti-apoptotic protein, was almost depleted in R5020-treated MCF-7PRB cells (Figure 4G). We reported that R5020 induced autophagy in these PRB-transfected MCF-7 cells [45]. Since BCL-2 inhibits autophagy, we assumed that the downregulation of BCL-2 in these cells would also cause an increase of autophagic activity. Indeed, R5020 also upregulated both LC3-I and LC3-II proteins (Figure 4G). It is plausible, therefore, that the downregulation of anti-apoptotic protein BCL-2 is important for R5020-induced cell death, but the mechanisms of the downregulations of other pro-apoptotic proteins require further investigation.

### 3.7. R5020 Exerts Robust Anti-Estrogenic Effect in MCF-7PRB Cells

It is shown in Figure 1A that R5020 depleted the ERα protein in MCF-7PRB cells. We tested if this downregulation of ERα plays a part in PR inhibition of cell growth by replenishing ERα using adenoviral transduction. Figure 5A shows that ERα protein in MCF-7PRB2 cells transduced with control adenovirus is undetectable after R5020 treatment. Transduction with adenoviral vector containing ERα cDNA increased the ERα level to ~3–4 times of the control transduction without R5020 treatment. However, ERα replenishment had no effect on R5020-induced growth inhibition (Figure 5B), suggesting that the downregulation of ERα could be responsible for R5020-induced growth inhibition.

Instead, R5020-induced downregulation of ERα was found to be associated with a strong anti-estrogenic effect on MCF-7PRB cells. In the absence of R5020, E2 stimulated cell growth in both MCF-7C and MCF-7PRB cells (Figure 5C). However, R5020 completely abolished E2-induced cell growth in MCF-7PRB cells in a 2D culture. MCF-7PRB cells were unable to survive in 2% Matrigel in the presence of R5020 with or without E2 treatment (Figure 5D). In contrast, MCF-7C cells grew well in R5020 or the R5020+E2-containing medium. Another unexpected observation was that vehicle-treated cells in 2% Matrigel grew better than the E2-treated cells. This suggests that some factors in Matrigel antagonized the effect of E2. In the tumorsphere culture, E2-treated MCF-7PRB cells formed thriving mammospheres, and the addition of R5020 completely abolished tumorsphere formation (Figure 5E).

### 3.8. R5020 Suppress the Genomic Effect of Estrogen through Inhibiting the Expression of ERα and Its Pioneer Factor FOXA1

FOXA1 is bona fide pioneer factor for ERα-chromatin interaction [46,47]. We found that R5020 treatment for 3 h significantly reduces the gene expression of both *ERα* and FOXA1 in MCF-7PRB cells but has no effect on MCF-7C cells (Figure 6A). Notably, the 3 h treatment of R5020 reduced FOXA1 expression by an average of 12 folds, suggesting that FOXA1 is a direct target gene of PRB. This notion is supported by the observation that 6 h treatment with cycloheximide (CHX), a protein translation inhibitor, did not abolish the repressive effect of R5020 on FOXA1 expression (Figure 6B). This downregulation of transcription was reflected at the protein level. R5020 markedly reduced both ERα and FOXA1 proteins (Figure 6C). A combined treatment of R5020 and E2 reduced ERα further compared to R5020 treatment alone because E2 has been known to down-regulate the ERα protein. The effect was sustained, whereby the downregulation of both proteins was more distinct at 96 h than at 24 h after treatment. This sustained downregulation of both ERα and FOXA1 by R5020 ensures a robust anti-estrogenic action.

To determine if the downregulation of ERα and FOXA1 proteins causes reduced chromatin-association of these proteins, a cell fractionation experiment was conducted to measure ERα and FOXA1 levels in cytoplasmic, nuclear, and chromatin-bound fractionations. Figure 6D shows that R5020 reduced ERα protein in all three fractions and chromatin-bound ERα was reduced by an estimated 70-80% based on the band intensity at both the 24 h and 96 h time point. Compared to ERα, which was present more abundantly in the cytoplasmic fraction, most of the FOXA1 protein was chromatin-bound (Figure 6D). At 24 hours’ and 96 hours’ after R5020 treatment, the chromatin-bound FOXA1 was reduced by 50% and 70%, respectively. As a loading control, the level and chromatin association of EZH2 were not affected by R5020. Interestingly, the nuclear- and chromatin-bound FOXA1 band was clearly upshifted compared to the cytoplasmic FOXA1. This could be due to the presence of a post-translational modification such as phosphorylation, which is necessary for the nuclear translocation of FOXA1. R5020-induced decrease of chromatin-bound ERα and FOXA1 led to a stark impairment of estrogen regulation of transcription. R5020 treatment for 24 h completely abolished E2-induced expression of *pS2* and *Areg*, two bona fide ERα target genes in MCF-7PRB cells (Figure 6E). R5020 also reduced E2-induced expression of *pS2* and *Areg* in MCF-7C cells to a small degree. This is likely mediated by the endogenous PR.

## 4. Discussion

### 4.1. Progestin Exerts Strong Anti-Tumoral Effects in MCF-7 Cells with High Levels of Stably Transfected PRB

Progesterone/progestin is unarguably the most controversial hormone with regards to its effect on breast cancer development. Laboratory and clinical studies have reported both pro-tumoral and anti-tumoral effect. This study provided unequivocal evidence that progestin is strongly anti-tumoral in MCF-7 cells with high levels of PRB. A single dose of R5020 induced irreversible replicative senescence (Figure 1). The observation agrees with studies in T47D and MDA-MB-231 cells expressing high levels of PRB, in which R5020-induced growth arrest [15,17,18] was observed. It appears that a sufficiently high level of PRB is required to mediate the anti-tumoral effect because studies on MCF-7 cells with much lower levels of stably transfected PR showed no detectable effect on cell growth in response to R5020 [26,27]. We speculate that a high level of PRB offers competitive advantages for the binding of transcription co-factors in order to exert its genomic effect [48]. The study suggests that high levels of PRB predict anti-tumoral effect of progestin in breast cancer.

PR/Progestin has also been shown to expand cell populations with cancer stem cell features. Using a CK5 reporter system, progestin was found to increase the population of CK5+ and CD44+ cells in the T47D and MCF-7 cell models [43]. Downregulations of microRNA (miR)-29a and miR-141 were reported to mediate progestin-induced expansion of breast cancer stem cells [49,50]. In addition, PR isoforms were reported to exert distinct influence on cancer cell stemness and proliferation potential. In the presence or absence of progestin, PRA-expressing T47D cells formed more tumorspheres, whereas PRB cells formed fewer but larger tumorspheres, suggesting that PRA could promote cancer cell stemness and PRB drives cell proliferation [50]. We report that R5020-treated cells were unable to form tumorspheres in the MCF-7PRB cells although it increased the CD44+/CD24+ populations (Figure 4). Furthermore, R5020-treated cells underwent enormous apoptosis when plated in Matrigel. This is consistent with reports that CD44+/CD24+ are generally non-tumorigenic, whereas CD44+/CD24-/low cells show cancer stem cell properties [51,52]. Thus, progestin did not promote cancer stem cell properties in MCF-7PRB cells.

### 4.2. The Mechanisms for High Levels of PRB-Mediated Growth Arrest Are Multifaceted

Aside from marked downregulation of various cyclins, CDK2 and CDK4 (Figure 2A), R5020 induced the activation of NF-κB subunit p65 (Figure 3B). This was associated with drastic up-regulation of NF-κB target genes *IL-1α*, *IL-1β* (Figure 3A)*,* which are important senescence-associated cytokines [53,54]. NF-κB is a master regulator of SASP [53,54]. Recombinant IL-1α and IL-1β have been reported to significantly inhibit the growth of MCF-7 cells [55,56]. Therefore, NF-κB activation is likely an important player in R5020-induced replicative senescence. Thus, the inhibition of cell cycle machinery and induction of SASP are likely important mechanisms of R5020-induced growth inhibition.

Cellular senescence and SASP are considered to be a double-edged sword. Increased levels of proinflammatory cytokines and chemokines could enhance the immune surveillance of senescent cells and elicit paracrine senescence to suppress the adjacent tumor cells [57,58]. SASP is also known to exert tumor-promoting effects on neighboring tumor cells [35]. Since R5020-treated MCF-7PRB cells were unable to form tumorspheres, it is unlikely to promote the pro-tumoral potential of neighboring cells. Oddly, R5020-induced apoptosis was associated with marked downregulation of pro-apoptotic markers including p53, BID, BAX, cleaved PARP, and Caspases 7 and 8, as well as depletion of the anti-apoptotic protein BCL-2 (Figure 4G). BCL-2 exerts an anti-apoptotic effect through the inhibition of BAX and BAK [59]. BCL-2 also inhibits autophagy by binding to Beclin 1, thereby disrupting Beclin-Vps34 interaction [60]. In two independent reports, BCL-2 gene silencing in MCF-7 cells has been shown to cause both apoptosis and autophagic cell death [61,62]. Consistently, the downregulation of BCL-2 by R5020 is associated with the induction of both apoptosis and autophagy, with the latter having been characterized in a previous study [63]. It has been reported in *Drosophila* models that autophagic cell death could lead to DNA fragmentation as a result of the autophagic degradation of the anti-apoptotic protein dBruce [64], mammalian orthologue of Bruce, or BIRC6. It is plausible that the huge downregulation of BCL-2 by R5020 is involved in both apoptosis and autophagic cell death.

### 4.3. R5020 Also Induces Molecular Changes That Promote Survival

Despite the profound growth-inhibitory effect on MCF-7 cells expressing high levels of PRB, R5020 also induced molecular changes that indicate pro-growth and pro-survival potentials in MCF-7PRB cells. First, similar to the biphasic effect reported in T47D cells [15,16], R5020 induced increases of cyclins A2, B1, and S-phase progression after 24 h treatment; however, prolonged exposure with R5020 reduced these levels. Second, R5020 markedly enhanced the protein levels of cyclin E consistently at both 24 h and 96 h of treatment, although this is not associated with induction of classical E2F target gene *c-Myc*. We speculate that R5020-induced downregulation of cyclin E-dependent CDK2 compromised the activation of cyclin E (Figure 2). Third, R5020 also significantly upregulated the expression of growth factor *Areg* (Figure 6E), which is a member of the EGF family and a major mediator of progesterone stimulation of mammary ductal growth [65]. *Areg* is also expressed at high levels in ER-positive breast cancers and was required for estrogen-dependent growth of MCF-7 tumor xenografts [66]. The conflicting properties of PRB are also illustrated in the regulation of apoptosis-related molecules, as has been described above (Figure 4G). It is thus clear that PRB is inherently capable of eliciting both pro- and anti-tumoral molecular changes, although the anti-tumoral activities are predominant in MCF-7 cells with high levels of PRB.

### 4.4. High Levels of PRB Mediates the Radical Anti-Estrogenic Effect by Down-Regulating ERα and FOXA1

Progestin is known to exert an anti-estrogenic effect in breast cancer cells. Several mechanisms have been suggested for the anti-estrogenic effect. PR has been reported to reprogram ERα cistromes so as to inhibit the genomic action of ERα [24,67]. It has also been reported that some of the ERα and PR target enhancers are organized into hormone-control regions (HCRs), in which the binding of ERα and PR in response to estrogen and progesterone leads to the opposite direction of gene regulation [68]. Additionally, FGF7/FGFR2 signaling was found to abolish the anti-estrogenic effect of progesterone in breast cancer cells [69]. The present study revealed an additional mechanism for the anti-estrogenic action of progestin. R5020 strongly down-regulated the expression of FOXA1 (12-fold after 3 h) and ERα, leading to a marked decrease in chromatin-bound FOXA1 and ERα (Figure 6). FOXA1 normally binds to chromatin to induce nucleosome rearrangement in order to create gene accessibility for specific transcription factors, such as ERα [70]. FOXA1 gene silencing blocks ERα-chromatin interaction and estrogen-induced gene expression [46]. Furthermore, FOXA1 form a transcriptional network with ERα and some other factors to control the susceptibility genes in breast cancer [71]. With reduced chromatin-bound FOXA1 and ERα as well as other mechanisms in operation, R5020 was able to elicit a complete blockade of estrogen-induced cell growth, tumorsphere formation and target gene expression in MCF-7PRB cells. Thus, the inhibition of FOXA1 and ERα expression could represent another major mechanism for the PR-mediated anti-estrogenic effect in breast cancer cells.

## 5. Conclusions

In summary, this study demonstrated unequivocally that progestin R5020 induces replicative senescence and apoptosis in MCF-7 cells with high levels of stable PRB expression. The effect was mediated by a myriad of molecular events, including downregulations of cyclins and CDKs, and activation of NF-κB signaling. Progestin also exerts a radical anti-estrogenic effect via drastic downregulation of ERα and FOXA1 and their chromatin association. Although the study used MCF-7 cells stably transfected with PRB, it delivered a clear message that progestin is strongly anti-tumoral and anti-estrogenic in breast cancer cells with high levels of PRB. This is supported by earlier reports that sustained progestin exposure in T47D cells and MB-MDA-231 cells with high levels of PR caused cell cycle arrest [15,16,17]. There is also clinical evidence that breast cancers with high levels of PR exhibit significantly better prognoses than those with low PR [72]. We propose that specific PR agonists should be evaluated in patient-derived xenograft models for their utility in endocrine therapy targeting breast cancer with high levels of PRB.

## Figures and Tables

**Figure 1 biomedicines-10-01860-f001:**
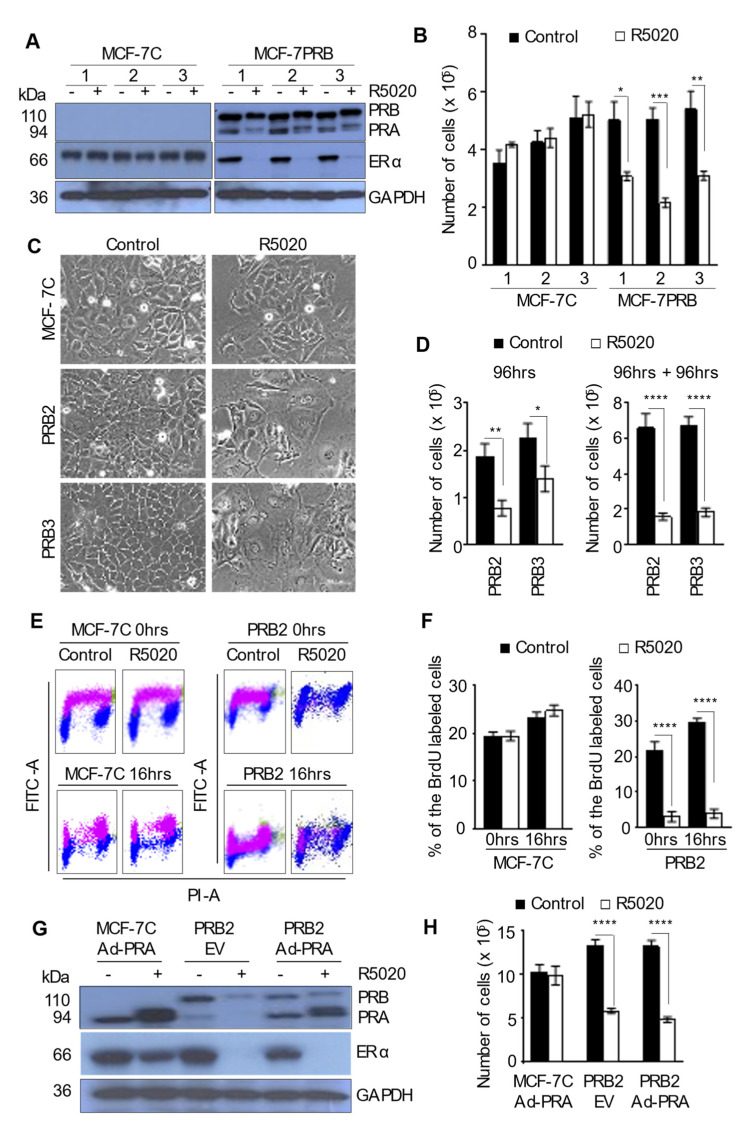
**R5020 induces irreversible cell cycle arrest in MCF-7 cells with PRB overexpression.** (**A**) Western blotting analysis of PR and ERα protein in stably transfected MCF-7 cells. Cell lysates of the three clones of vector-transfected (MCF-7C) and PRB expression vector-transfected cells (MCF-7PRB) were collected after treatment with vehicle or 10 nM R5020 for 24 h and probed with anti-PR, anti-ERα, and anti-GAPDH. GAPDH levels were used as the loading control. (**B**) Cell growth of MCF-7C and MCF-7PRB in response to a single dose of 10 nM R5020 for 96 h. Cell numbers were quantified by counting. (**C**) R5020 induced cell spreading in MCF-7PRB cells. Cell images were taken 96 h post-treatment. (**D**) MCF-7PRB cells remain growth-arrested at 96 h after R5020 removal. Left panel, cell numbers at 96 h after R5020 treatment. Right Panel, cell numbers at 96 h after R5020 withdrawal following 96 h treatment. (**E**,**F**) R5020 blocked DNA replication. Cells were treated with either vehicle or 10 nM R5020 for 96 h before they were pulse-labeled with 20 μM BrdU. At the indicated time, the cells were stained with the FITC-labeled anti-BrdU antibody and Propidium iodide (PI) and analyzed by flow cytometer. (**E**) Representative dot plot of BrdU incorporation at 0 h and 16 h post-BrdU-labelling. (**F**) Percentage of BrdU-labelled cells with and without R5020 treatment. (**G**,**H**) PRA had no effect on cell growth in response to R5020. MCF-7C and MCF-7PRB2 were transduced with Ad-EV (control vector), or Ad-PRA vectors for 24 h prior to treatment. (**G**) Western blotting analysis of PRA, PRB, and ERα after 24 h treatment. (**H**) Cell numbers of MCF-7C and MCF-7PRB2 were determined by counting. All numeric data are presented as the mean ± SEM, *n* = 9 from 3 independent experiments. Asterisks indicate statistical significance between control and R5020 treated samples (* *p* < 0.05, ** *p* < 0.01, *** *p* < 0.001, **** *p* < 0.001).

**Figure 2 biomedicines-10-01860-f002:**
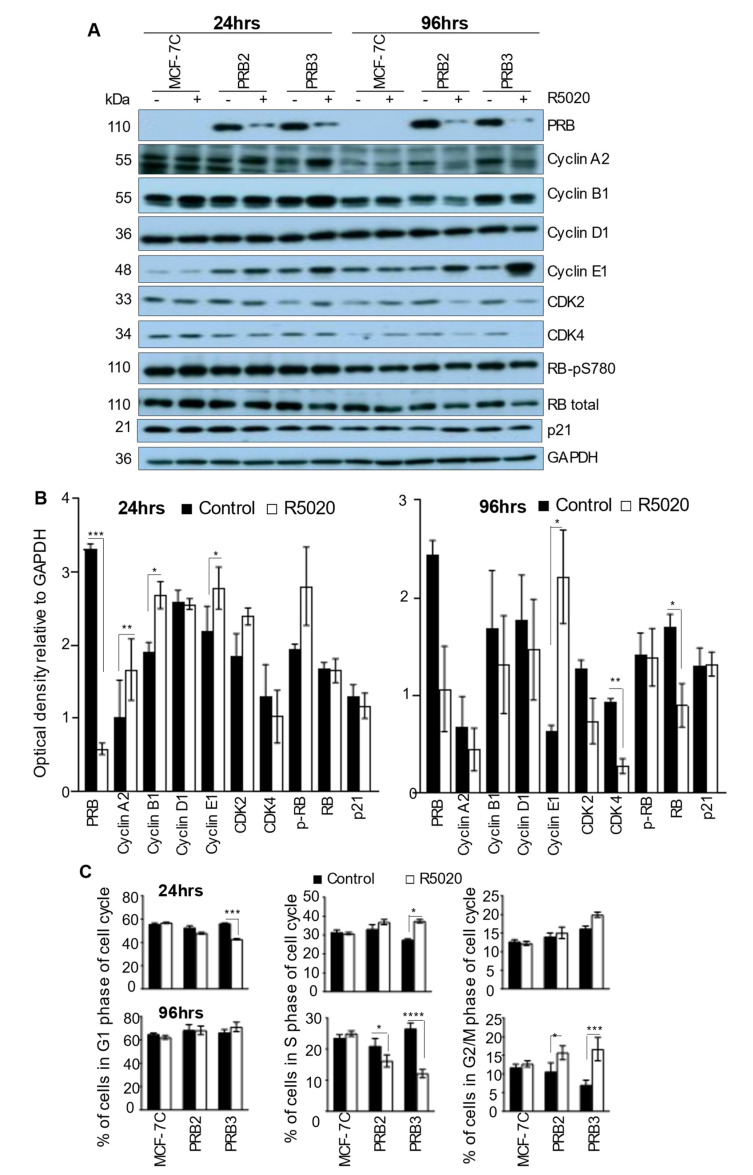
**R5020 exerts a biphasic effect on cell cycle regulators and cell cycle progression in MCF-7PRB cells.** (**A**) Protein levels of various cell cycle regulators were analysed by Western Blotting analysis after 24 h and 96 h treatment with R5020. GAPDH was probed as loading control. (**B**) Densitometry analysis of Western blots in (**A**) The optical density was quantitated by image J. The results are expressed relative to the intensity of GAPDH for each sample. The data are from two independent experiments with values from PRB2 and PRB3 considered as individual observations, giving rise to *n* = 4 in each group. The results are expressed as mean ± SEM. GAPDH was used as a loading control for normalization. (**C**) R5020 enhanced S-phase cells after 24 h but reduced it after 96 h in MCF-7PRB cells. The cell cycle distribution of the individual clones of the MCF-7C and MCF-7PRB cells were determined by flow cytometry analysis of PI staining. Data are presented as the percentage of the cells in the G1, S, and G2/M phase of the cell cycle (mean ± SEM, *n* = 9 from 3 independent experiments). The degree of statistical significance is indicated with asterisks (* *p* < 0.05, ** *p* < 0.01, *** *p* < 0.001, **** *p* < 0.001).

**Figure 3 biomedicines-10-01860-f003:**
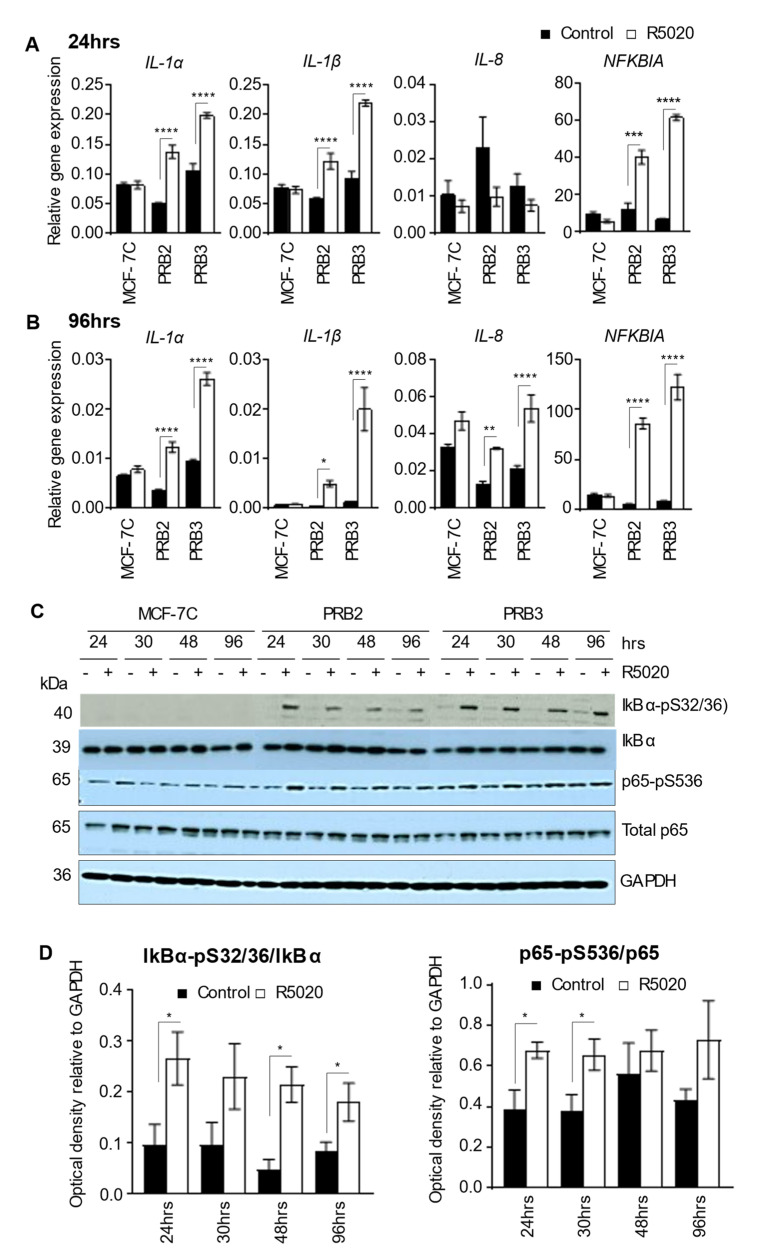
**R5020 activates NF-κB pathway and SASP in MCF-7PRB cells.** (**A**) Relative gene expression of *IL-1α*, *IL-1β*, *IL-8,* and *NFKBIA* at 24 h after R5020 treatment. (**B**) Relative gene expression of *IL-1α*, *IL-1β*, *IL-8,* and *NFKBIA* at 96 h after treatment. The data are representative of three independent experiments (mean ± SEM, *n* = 9). (**C**) Protein levels of phosphorylated and total IkBα and p65 over a 96-h period were analyzed by Western blotting. GAPDH was probed as a loading control. (**D**) R5020 treatment increased the ratio of IkBα-pS32/36/IkBα and p65-pS536/p65. The optical density was obtained by image J. The results are normalized to the intensity of GAPDH for each sample. The data are from two independent experiments with values from PRB2 and PRB3 considered as individual observations, giving rise to *n* = 4 in each group. The results are expressed as mean ± SEM. Asterisks indicate the statistical significance between R5020 against control (* *p* < 0.05, ** *p* < 0.01, *** *p* < 0.001, **** *p* < 0.001.

**Figure 4 biomedicines-10-01860-f004:**
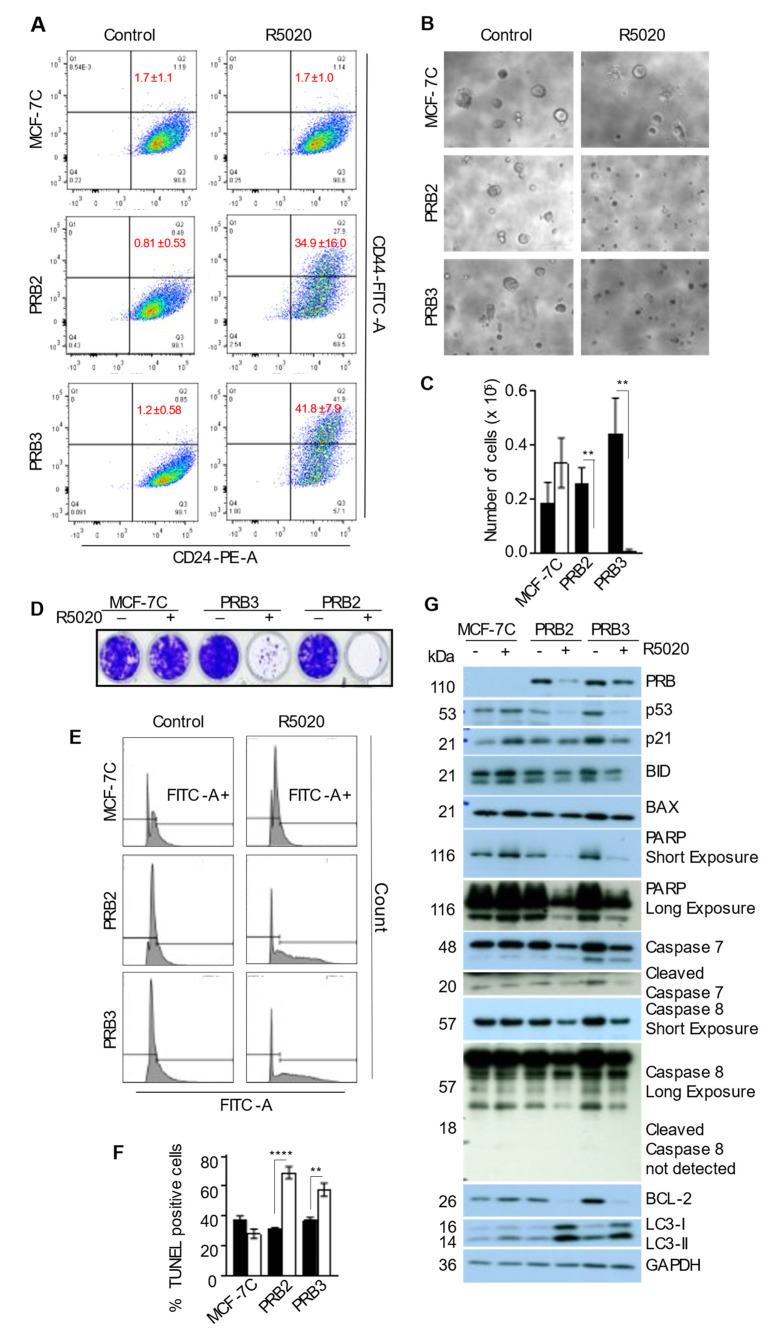
**R5020-treated MCF-7PRB cells were unable to form tumorspheres and underwent massive apoptosis.** (**A**) R5020 increased CD24+CD44+ cell population in MCF-7PRB cells. Cells were treated with R5020 for 96 h before being stained with anti-CD44-PE and anti-CD24-FITC antibodies for Flow cytometric analysis. Quadrants were set using appropriate controls. The numbers in Q2 are the average percentage of CD24+CD44+ cells from three independent experiments (Mean ± SEM, *n* = 9). (**B**) R5020-treated cells were unable to form tumorspheres. Cells were treated with 10 nM R5020 for 72 h before being plated onto Matrigel for tumorsphere growth, and the images were taken on day 14 after plating. (**C**) R5020-treated cells did not survive the tumorsphere culture. Cells from B. were released from Matrigel by dispase and trypsin digestion before viable cells were counted after staining with trypan blue. (**D**) R5020-treated MCF-7PRB cells did not survive when plated in medium containing 2% Matrigel. Cells were treated with 10 nM R5020 for 72 h before being plated in medium containing 2% Matrigel and 10 nM R5020. The cells were cultured in their respective treatment for 14 days before they were stained with 0.1% crystal violet. (**E**) R5020-treated MCF-7PRB cells underwent apoptosis, as analyzed by a TUNEL assay. MCF-7PRB cells were treated with 10 nM R5020 for 72 h before being plated in 2% Matrigel. After 48 h, DNA fragmentation was evaluated by a FACS-based TUNEL assay. The FITC-A- indicates the percentage of cells with undamaged DNA. FITC-A+ indicates the percentage of cells with fragmented DNA. (**F**) The percentage of FITC-positive cells are presented as the mean ± SEM, *n* = 9 from 3 independent experiments. (**G**) Analysis of the apoptosis-related proteins by Western blotting analysis. Cells were treated with 10 nM R5020 for 72 h before being seeded in 2% Matrigel and cell lysates were collected for Western blotting analysis at 48 h plating. GAPDH was analyzed as loading control. The degree of statistical significance is indicated with asterisks (** *p* < 0.01, **** *p* < 0.001).

**Figure 5 biomedicines-10-01860-f005:**
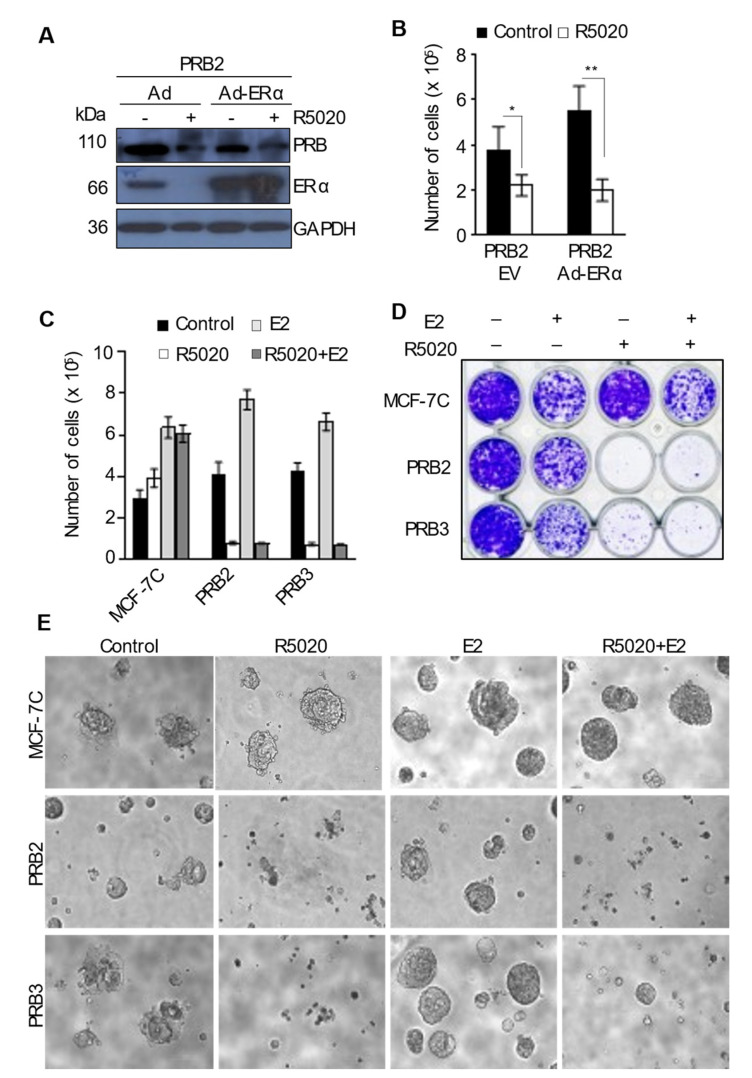
**R5020 abolished estrogen-induced cell growth and tumorsphere formation.** (**A**,**B**) cells were plated in 5% DCC-FCS medium and transduced with adenoviral control vector (Ad) or Ad-ERα. A total of 24 h after the transduction, cells were treated with 10 nM R5020 for 96 h. (**A**) Western blotting analysis of PR and ERα protein in Ad and Ad-ERα-transduced cells, and GAPDH was analyzed as the loading control. (**B**) ERα overexpression did not affect R5020-induced growth inhibition in MCF-7PRB cells. (**C**) R5020 abolished estrogen-induced growth in MCF-7-PRB cells after treatment for 96 h. R5020 treated cells are unable to survive in 2% Matrigel (**D**) or to form tumorspheres with or without E2 treatment (**E**). All numeric data are presented as the mean ± SEM, *n* = 9 from 3 independent experiments. The degree of statistical significance between control and R5020 treatment is indicated by Asterisks (* *p* < 0.05, ** *p* < 0.01).

**Figure 6 biomedicines-10-01860-f006:**
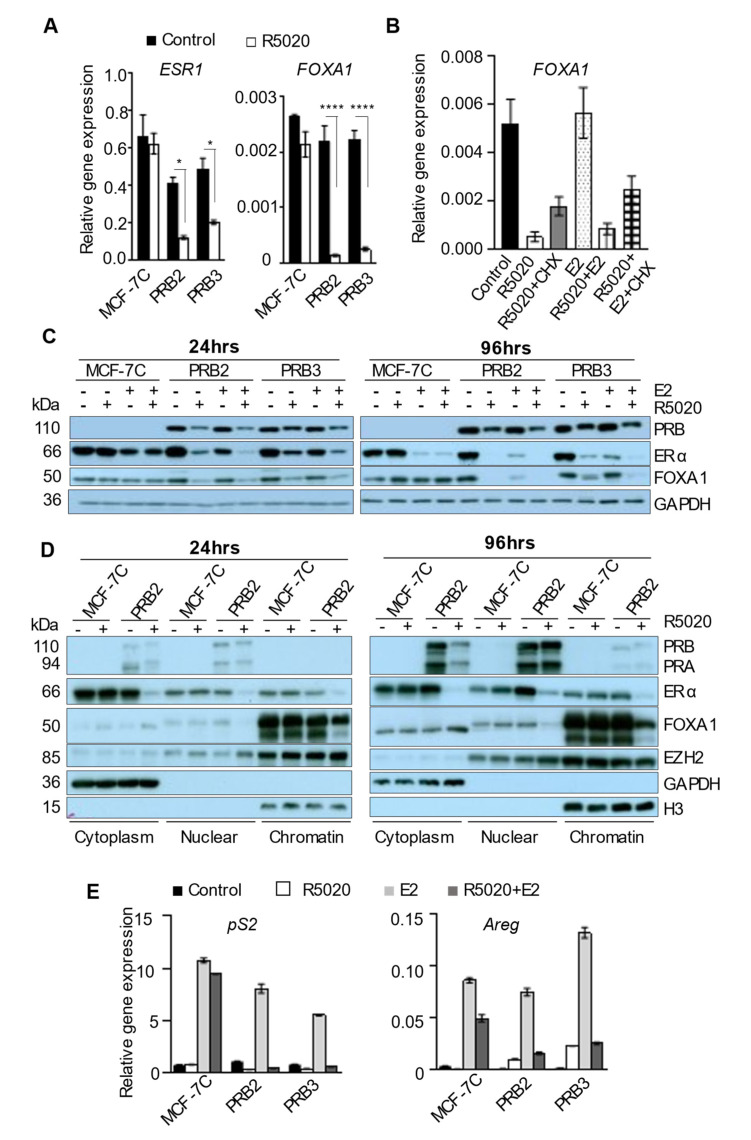
**R5020 represses the expression of****ERα and FOXA1 and their chromatin association.** (**A**) R5020 treatment of MCF-7PRB cells for 3 h significantly down-regulated the expression of *ESR1 and* FOXA1. (**B**) Treatment with cycloheximide (CHX) for 6 h was unable to abolish R5020 repression of FOXA1 gene expression. (**C**) Treatment with R5020 markedly reduced protein levels of ERα and FOXA1 in MCF-7PRB cells, as determined by Western blotting analysis. GAPDH was probed as a loading control. (**D**) R5020 diminished chromatin-bound ERα and FOXA1 in a cell fractionation assay. GAPDH, EZH2, and H3 were used as markers for cytoplasmic, nuclear, and chromatin-bound fractions, respectively. (**E**) R5020 treatment for 24 h abolished estrogen-induced expression of *pS2* and *AREG* in MCF-7PRB cells. All numeric data are presented from three independent experiments (mean ± SEM, *n* = 9). The degree of statistical significance is indicated with asterisks (* *p* < 0.05, **** *p* < 0.001).

**Table 1 biomedicines-10-01860-t001:** Primers used for qRT-PCR.

Primer Name		Primer Sequence–5’ to 3’
*AREG*	Forward	GTGGTGCTGTCGCTCTTGATA
	Reverse	CCCCAGAAAATGGTTCACGCT
*ERα*	Forward	GAGGGCAGGGGTGAA
	Reverse	GGCCAGGCTGTTCTTCTTAG
FOXA1	Forward	GCAATACTCGCCTTACGGCT
	Reverse	TACACACCTTGGTAGTACGCC
*IL-1α*	Forward	AGATGCCTGAGATACCCAAAACC
	Reverse	CCAAGCACACCCAGTAGTCT
*IL1β*	Forward	TTCGACACATGGGATAACGAGG
	Reverse	TTTTTGCTGTGAGTCCCGGAG
*IL-8*	Forward	GGGCCAAGAGAATATCCGAAC
	Reverse	TGGATCCTGGCTAGCAGACTA
*NFKBIA*	Forward	CTCCGAGACTTTCGAGGAAATAC
	Reverse	GCCATTGTAGTTGGTAGCCTTCA
*pS2*	Forward	GCCATCGAGAACACTCAAGAAGAAG
	Reverse	ACTGTGTCACCAGCCAGATGGA
*36B4*	Forward	GATTGGCTACCCAACTGTTGCA
	Reverse	CAGGGGCAGCAGCCACAAAGGC

## Data Availability

Data are contained within the article in the Appendix A.

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
