# Peer review of "High Levels of Progesterone Receptor B in MCF-7 Cells Enable Radical Anti-Tumoral and Anti-Estrogenic Effect of Progestin"

_biomedicines, 2022, doi:10.3390/biomedicines10081860_

Round 1

Reviewer 1 Report

The manuscript entitled: Overexpression of progesterone receptor isoform B in MCF-7 cells enables progestin’s radical anti-tumoral and anti-estrogenic effect documents the molecular mechanism responsible for the strong antiestrogenic effect of PR agonist. The team has extensively investigated the expression of high levels of PR in MCF-7 cells, enabling a strong anti-tumoral response to progestin. The manuscript is exciting and scientifically sound; however, some significant concerns must be addressed. Please find my comments below.

Overall, the manuscript is too wordy, and it should be shortened. There are a few grammatical errors that should be corrected. Some western blots are over-exposed/supersaturated and need better quality images. 

The abstract should be shortened with the essential information, including the hypothesis and findings. I suggest removing Line #21 to Line #24, or please rephrase it to make it short.

Nuclear factor NF-kB should be represented either as Nuclear factor-kB or NF-kB.

Figure #1 Panel A and G, GAPDH blots are supersaturated and need a better quality image. Panel D/F/H the lines comparing the bars are misplaced. Please rearrange them for better representation.

Figure 6: The basal level of ERa expression in MCF-7C is significantly high than that observed in Figure #1 panel G. I struggled to understand Why? 

Figure 3_Panel C: What is the significance of including 30 hours time point between regular time intervals of 24 hours. Adding a line or two in methods or discussion will be helpful. 

Figure3C_Panel C: The quality of the figure needs to improve even in the supplementary/full blot images. 

Most of the figures have different intensities of bands of loading controls. Also, Figure 4 Panel G and Figure 6 Panel C has an enormous scope of improvement in the GAPDH blots. What was the amount of protein loaded initially in the Gel? 

The authors should add some recent references. I highly encourage citing recent year references. The latest reference that has been added is a single reference for the 2020 NCI workshop report. 
I strongly encourage authors the include densitometric analysis of western blots in the manuscript.

Reviewer 2 Report

In this study authors evaluated whether MCF-7 cells engineered to overexpress PRB by stable transfection were sensitive to progestin and found that a single dose of progestin R5020  induced an irreversible cell growth arrest and senescent-associated secretory phenotype in MCF-7 cells with PRB overexpression but had no effect on cells with PRA overexpression. R5020 also induced an activation of nuclear factor NF-κB and upregulation of interleukins IL-1α, IL-1β and IL-8. Moreover, R5020-activated PRB abolished the effect of estrogen downregulating ERα and FOXA1. 

This study is very interesting and generally well written but some points need to be improved. In particular: 

Lines 46-60 : It deserves to be pointed out that progesterone receptor expression has also been associated with improved overall survival and progression-free survival in ovarian cancer (another important gynaecological cancer) where the expression PRG and ER is tightly regulated by NRF2/KEAP1 pathway, a key modulator of cell proliferation, migration and chemoresistance onset in cancer cells (as recently reviewed PMID: 35453348). This is a very interesting point to highlight because hormone replacement therapy and progestin sensitivity may exert the same function in this type of cancer adding further importance to the results found by the authors.

Line 67: Authors must specify that  R5020 is progestin (promegestone)

Protein Lysate collection and Western Blotting analysis: primary antibodies product codes and dilutions used must be reported 

Western blots: In all figures where western  blots are shown, molecular weights must always be reported

Figure 2A: Have densitometric analysis been done?  Densitometric analysis must be shown

Figure 2B and 3C: Rappresentative Histograms of Cell Cycle Phases should be reported

References: Authors must follow the journal style

Round 2

Reviewer 1 Report

The authors have made considerable changes to the manuscript, and the quality of the manuscript is improved significantly. 

Line #301: LOT number is missing. 

In the "List of Abbreviations," all the text should be uniform.

Round 3

Reviewer 2 Report

the manuscript has been significantly improved and can be accepted in the present form.